# Current Immune Checkpoint Inhibitor Genetic Biomarker Exploration in Gastrointestinal Tumors

**DOI:** 10.3390/cancers14194804

**Published:** 2022-09-30

**Authors:** Jane E. Rogers, Kohei Yamashita, Matheus Sewastjanow Silva, Jaffer A. Ajani

**Affiliations:** 1U.T. M.D. Anderson Cancer Center Pharmacy Clinical Programs, Houston, TX 77030, USA; 2U.T. M.D. Anderson Cancer Center Department of Gastrointestinal Medical Oncology, Houston, TX 77030, USA

**Keywords:** gastrointestinal neoplasms, immune checkpoint inhibitor, microsatellite-instability-high/deficient mismatch repair, programmed-death-1, programmed death-ligand-1

## Abstract

**Simple Summary:**

Immune checkpoint inhibitors (ICIs) are now incorporated into the management of GI tumors. The heterogenous nature of these tumors, however, reveals a lack of ICI consistency in effectiveness. Certain biomarkers have emerged as being potentially predictive for ICI effectiveness. Our review focuses on these biomarkers while discussing the current limitations with these markers.

**Abstract:**

Immune checkpoint inhibitors have revolutionized cancer management. Some patients with gastrointestinal (GI) tract malignancy have experienced remarkable results. Here, in our review, we discuss predictive/prognostic GI tumor biomarkers that appear to correlate with benefits with this strategy. Remarkable progress has been made in certain subsets of patients including the potential for solid tumor patients to avoid local therapies such as radiation and/or surgery (organ preservation), which come with acute and chronic risks that have historically been the only curable strategies for these GI tumors. These results provide new and exciting strategies for solid tumor management. Unfortunately, immune checkpoint inhibitors can correlate with biomarkers, but benefits occur in a small subset of patients with GI malignancies. Most frequently, immune checkpoint inhibitors fail to induce response in GI malignancies due to the “cold” tumor microenvironment that protects cancer. Translational strategies are needed to develop effective combination strategies and novel biomarkers to overcome the intrinsic resistance.

## 1. Introduction

Gastrointestinal (GI) tumors, cancers occurring in the digestive tract, encompass an array of heterogeneous solid tumors. GI tumors consist of some of the most commonly diagnosed malignancies (i.e., colon cancer, rectal cancer, pancreatic cancer, hepatocellular carcinoma, gastric cancer, and esophageal cancer) and encompass some rare tumor entities (i.e., anal, biliary tract cancers, gallbladder, appendiceal, duodenal, etc.). These tumors differ considerably in their risk factors, location, histological characteristics, molecular profile, and management. Additionally, each tumor type has many heterogeneous subtypes.

Molecular profiling has expanded our understanding in identifying targets and predictive biomarkers. This is true for GI tumors. Like other solid tumors, immune checkpoint inhibitors (ICIs) are now being incorporated into treatment; however, not all patients respond similarly. Continued exploration of biomarkers remains of utmost importance to determine the best ICI precision medicine in GI tumors. Here, in our review, we discuss some of the steps taken in ICI biomarkers and their relationship with GI tumor treatment.

## 2. Microsatellite Instability-High/Deficient Mismatch Repair (MSI-H/dMMR)

DNA MMR machinery is essential for the maintenance of genomic stability. The MMR machinery is composed of MSH2/MSH6 and MSH2/MSH3 that recognize single-nucleotide mismatches and small insertion/deletions that occurs during DNA replication. Subsequently hMLH1/hPMS1 Homolog 2, (hPMS2), hMLH1/hPMS1 Homolog 1(hPMS1) and hMLH1/hMLH3 are recruited to catalyze the excision and resynthesize the mismatch [1]. The dysfunction of this system, namely dMMR, results in an errors in microsatellites which consist of repeated DNA sequences of 1–6 nucleotides [2]. Thus, the alteration of the number of microsatellites sensitizes a dMMR state. This is referred to as microsatellite instability (MSI). MSI is thought to be involved in tumorigenesis and tumor proliferation due to the accumulation of repair-associated mutations in genes for tumor suppression, cell proliferation, DNA repair, apoptosis. Clinically, it can be categorized as MSI-H and MSI-low or stable (MSS) according to the frequency of MSI [3]. In sporadic dMMR tumor, it is mainly caused by MMR gene mutation due to acquired hypermethylation of the promoter region of *MLH1* gene, leading to decreased expression of MMR protein [4]. On the other hand, the Lynch syndrome, which takes the form of autosomal dominant inheritance, is caused by germline mutations of the MMR-regulated genes (*MLH1*, *MSH2*, *MSH6*, and *PMS2*) or a deletion of the *EPCAM* gene adjacent to the upstream of the *MSH2* in one allele [5,6,7].

MSI-H/dMMR solid tumors are found in various organs [8,9]. The frequency of MSI-H/dMMR in colorectal cancer (CRC) is reported to be approximately 15% [10] with Lynch syndrome-associated CRC accounting for ~20–30% of cases and sporadic MSI-H/dMMR CRC being ~70–80% of cases [11]. The frequency of MSI-H/dMMR CRC varies according to stage (~20% stage I/II, 12% stage III, and 5% in stage IV) [12]. MSI-H/dMMR CRC is more common in the right colon and the proportion of poorly differentiated adenocarcinoma is high [13]. Moreover, BRAF V600E gene mutation is found in 35–43% of MSI-H/dMMR CRC [14,15]. Since BRAF V600E gene mutations are rarely found in the Lynch syndrome-related CRC, BRAF screening in MSI-H/dMMR CRC helps to distinguish sporadic MSI-H/dMMR tumor or Lynch syndrome [16].

In gastric cancer (GAC), MSI-H/dMMR has a frequency of ~20% [17,18]. As well as MSI-H/dMMR CRC, the prevalence depends on tumor stage; the highest in node-negative stage (20%) and the lowest in metastatic disease (<5%). In esophageal adenocarcinoma (EAC), MSI-H/dMMR can be observed 3–5% due to somatic mutation since Lynch syndrome associated esophageal adenocarcinoma (EAC) is rare [19]. Regarding gastroesophageal junction (GEJ) cancer, those in Siewert type II and III are related to MSI-H/dMMR [20]. In small bowel cancer, the frequency of MSI-H/dMMR is reported to be 5–45%, which is a relatively big range and frequency [21]. MSI-H/dMMR is also associated with other GI tumors at a lower incidence (i.e., ~2–2.5% pancreatic; ~2% biliary; ~2% gallbladder, etc.) and even more rarely seen in some (Hepatocellular carcinoma (HCC) and anal cancer given different cancer etiologies) [22,23,24].

MSI-H/dMMR tumors are generally associated with a high neoantigen burden, highly immunogenic, and thus thought to respond to ICI therapy. A current exciting pathway for these tumors is initial investigation with localized MSI-H/dMMR solid tumors and the potential for organ-sparing (non-operative) approach. Cercek et al. recently published results of locally advanced MSI-H/dMMR rectal cancer patients (*n* = 12) who received anti-programmed death-1 (anti-PD-1) agent ICI, dostarlimab [25]. All patients had a clinical complete response (cCR). Currently, no patients had received chemoradiation (CRT), undergone surgery, progressed, or had recurrence. Additionally, Ludford et al. reported initial results giving pembrolizumab to MSI-H/dMMR tissue agnostic localized primary tumors (*n* = 32) [26]. Tumor types included 24 CRC and 8 non-CRC (1 endometrial, 1 gastric, 1 meningeal, 2 duodenal, 1 ampullary, 2 pancreatic). Among 30 evaluable pts, overall response rate (ORR) was 77% with 30% CR, 47% PR, 20% stable disease, 3% progression. Pathological CR (pCR) was noted in 50% of the six patients that underwent surgery. An organ-sparing approach was chosen in 15 patients and two patients had reached one year of avoiding surgery. Additionally, an ICI shift in upfront treatment for advanced MSI-H/dMMR CRC has been a recent development [27]. These patients are now recommended pembrolizumab monotherapy given superiority over chemotherapy seen in KEYNOTE-177. In the KEYNOTE-158, a phase 2 pembrolizumab study, an ORR of 40.9%, a median progression-free survival (PFS) of 4.2 months, and median OS of 24.3 months was seen in advanced, pre-treated MSI-H/dMMR biliary tract cancers (*n* = 22) [28]. Although dMMR/MSI-H status for GI cancer has become a biomarker that determines an indication for ICIs, factors associated with resistance are still being investigated. Further research would suggest more implications for the role of dMMR/MSI-H status as a predictive marker for immunotherapy but determine why certain MSI-H/dMMR patients do not respond will be the key to moving forward. We believe tissue-agnostic trials of MSI-H/dMMR tumors will provide answers in a quicker fashion. Exciting trials are underway in both the localized MSI-H/dMMR rectal, colon, and gastric setting and metastatic solid tumor setting as these patients will need differing treatment strategies than those proficient in MMR/MSS. Phase 3 trials in this space are described in Table 1 [29,30,31,32,33,34,35].

## 3. Programmed-Death Ligand-1 (PD-L1) Expression

Given the role that PD-L1 plays in tumor immune escape, its expression has emerged as a potential biomarker to test the effectiveness of ICI. PD-L1 expression is a current exploration amongst GI tumors to determine if this holds an ICI predictive role.

For upper GI patients (gastric and esophageal patients), PD-L1 expression and ICI response is of much debate given conflicting results seen in CHECKMATE 649, KEYNOTE-811, ATTRACTION-4, JAVELIN, and ORIENT-16 [36,37,38,39,40]. These trials are described in detail in Table 2. Currently, for upper GI tumors, we feel PD-L1 expression (method and degree of positivity) needs more standardization across trial designs to determine the predictive value. It is clear with the current data that additional new biomarkers and correlating with other clinicopathological features are needed to determine those likely to benefit in the high PD-L1 combined positive score (CPS) patients. As this appears at present time not to be the ideal biomarker alone to determine ICI response.

PD-L1 expression does not appear currently to be key marker for biliary tract cancers as seen in the TOPAZ-1 trial in which the current standard practice is for upfront advanced tumors to receive durvalumab with gemcitabine plus cisplatin [41]. PD-L1 expression also did not appear to hold predictive value in the KEYNOTE-158 study [42]. ICI biomarker relevance will be needed soon; however, in advanced biliary tract tumors as results of the phase 3 gemcitabine plus cisplatin +/− nab-paclitaxel, results are expected soon [43]. Determining who would benefit most from ICIs might help determine the best upfront therapy if this trial’s results are favorable. Controversial results are noted for HCC. CHECKMATE-459, nivolumab compared to sorafenib, those with PD-L1 positive reported higher ORR [44]. While CHECKMATE-040 showed no statistical difference [45]. Of significance are updated results of the IMbrave150. IMbrave150 established atezolizumab and bevacizumab are standard front-line treatment for advanced HCC. An updated retrospective look at the tissues in this study showed that PD-L1 expression is likely of limited predictive value to determine benefit with atezolizumab and bevacizumab (median overall survival (OS) 12.6 months PD-L1 ≥1%; median OS 15.4 months for PD-L1 <1%) [46]. Additionally, results of the HIMALAYA trial of tremelimumab and durvalumab are expected to be added to the treatment choices for upfront HCC [47]. PD-L1 expression in relation to outcomes in HIMALAYA lacked reporting thus limit determination of an ICI predictive link.

For squamous cell carcinoma of the anal canal (SCCA), single agent ICIs (nivolumab or pembrolizumab) are options for refractory metastatic anal cancer [48,49]. PD-L1 expression was not required in the studies evaluating the use of these agents in this refractory patient population; however, exploratory analysis suggests higher response in those with PD-L1 expression [48]. These data, however, remain too immature for any value.

There is much work needed at understanding the predictive role of PD-L1 expression regarding ICI GI therapy as currently it has not been as precise as hoped. Questions remaining include (1) determining the standard definition for PD-L1 expression, (2) what tumor should be tested (fresh; archived) (3) why does expression not correlate to response (4) why do some non-PD-L1 expressing tumors shrink (5) is expression altered by prior therapy (6) does PD-L1 expression drive immunogenicity in the same fashion across tumors? For now, we believe PD-L1 expression correlation remains too vague for most GI tumors and continued exploration is needed to determine the role in each tumor type.

## 4. Tumor Mutation Burden (TMB)

Cancer is essentially a genetic disease. Genetic alterations, such as non-synonymous mutations, synonymous mutations, insertions or deletions, and copy number variation, can lead to carcinogenesis and cancer evolution. These mutations are transcribed and translated to produce neoantigen-containing peptides that are loaded onto major histocompatibility complex (MHC) molecules and presented on the surface of cancer cells for recognition by T cells [50]. Although not all mutations steadily produce neoantigens and thus are immunogenic, more somatic mutations are more likely to produce neoantigens and it supposed to enhance the immune surveillance [51]. In this regard, tumor mutation burden (TMB), which indicates a total number of genetic mutations in cancer, has emerged as a new promising predictive biomarker for ICI.

TMB is reported as the number of mutations per megabase (mut/Mb). Initially, TMB was determined using whole exome sequencing (WES) which included non-synonymous mutations in exomes of tumor DNA and excluded germline mutations by comparison with matched normal DNA. However, due to the high cost and technical complexity of WES, the comprehensive genetic panel by next generation sequencing (NGS) has come to be used as an alternative to TMB measurement in clinics. In fact, the comprehensive genomic profiling assay established by Foundation Medicine (F1CDx), can provide accurate assessment of TMB compared with those using WES in previous report [52]. This panel covers 324 genes including synonymous mutations, short indels, and splicing mutations as well as non-synonymous mutations. Another well-known NGS assay is the MSK-IMPACT developed at Memorial Sloan Kettering Cancer Center. This assay covers 468 cancer related-gene mutations, corresponding to 1.22 Mb. The US FDA approved both F1CDx and MSK-IMPACT as diagnostic tools for TMB in 2017. Moreover, the blood-based assay has also been used for TMB estimation in circulating tumor DNA. Some of the possible benefits of this technology include the ease of noninvasive sample collection and the ability to repeat sampling during therapy. Recent clinical trial demonstrated blood TMB (bTMB) could be a predictive biomarker for atezolizumab in non-small cell lung cancer [53]. More research is needed to determine the usefulness of bTMB in solid tumors, including GI cancers.

The Cancer Genome Atlas (TCGA) project has revealed a broad distribution of TMB across 20–30 cancer types using WES [54]. In this analysis, TMB was higher in cancers linked to chronic genotoxic exposure, such as UV light for melanoma and tobacco for lung cancer, while TMB was lower in leukemia and childhood cancers. According to an NGS-based analysis of a large patient cohort representing 14 distinct GI cancers, the highest TMB tumors were most common in right-sided colon and small-bowel adenocarcinomas (average TMB of 13 and 10.2 mutations/Mb, respectively), while the lowest frequencies of TMB-high tumors were seen in pancreatic adenocarcinoma and gastrointestinal stromal tumors (average TMB of 1.3 percent and 0 percent, respectively) [55]. Another integrated analysis across 24 cancer types using WES reported colorectal and stomach adenocarcinoma showed multimodal TMB distribution [56]. This study also reported the percentage of TMB greater than 10 mut/Mb were 29% in GAC, 18% in CRC, 9% in esophageal adenocarcinoma, respectively. Notably, most of those cancers exhibited MSI-H. As previously described, MSI-H tumor cause frameshift mutations, suggesting that MSI-H tumors have a high TMB. Indeed, Chalmers et al. reported approximately 83% of MSI-H tumor had TMB-H defined as 20 mut/Mb or higher, and 97% had TMB of 10 mut/Mb or higher [52]. Conversely, only 16% with high TMB were classified as MSI-H in all cancer types, suggesting MSI-H is just one of the factors leading to high TMB. However, it was shown that GI cancers such as gastric, duodenum, and small intestine adenocarcinoma often co-occurred MSI-H and TMB-H.

The concept of link between increased TMB and greater ICI response appears to be verified in various cancer types in multiple retrospective studies [57,58]. Goodman et al. reported tumor response rate and TMB level were linearly related in a pan-cancer investigation of 151 patients treated with anti–PD-1/PD-L1 monotherapy [59]. A strong correlation was also found with the dichotomized TMB level with cut off value of 20 mut/Mb, which was constant across tumor types. Furthermore, TMB was retrospectively analyzed in prospective clinical trial to explore its ability as a predictive biomarker of response in ICI. In KEYNOTE 158 trial, TMB-high (≥10 mut/MB) was associated with pembrolizumab efficacy in patients with previously treated unresectable or metastatic solid tumors, including GI cancer [60]. In this trial, 102 of 790 eligible patients exhibited MSI-H, with overall response rate of 29%. Notably, the higher clinical benefit in the TMB-high category was not entirely accompanied with MSI-H status. Based on these findings, the FDA granted accelerated approval for pembrolizumab monotherapy in patients with TMB-H (≥10 mut/MB) solid tumor utilizing the companion diagnostic assay (F1CDx) who are refractory to therapy and for whom alternative therapies are inadequate.

The utility of TMB as a predictive biomarker for ICIs in certain GI cancer types is currently being explored. Especially, TMB in CRC is often discussed in the context of MSI status. Within MSI-H metastatic CRC, TMB demonstrated to be an important independent biomarker for patient stratification based on ICI therapeutic efficacy [61]. In an exploratory analysis from KEYNOTE-061, a strong correlation between TMB and pembrolizumab efficacy was demonstrated in patients with GAC and GEJ adenocarcinoma, further suggesting that TMB may be a significant and independent predictor beyond PD-L1 status [62]. Further research is needed to elucidate the role of TMB as a biomarker in GI cancer. Currently, the Friends of Cancer Research TMB Harmonization Group is working on harmonization of TMB assays and detection of optimal cutoff values, which may also be involved in future biomarker research of TMB in GI cancer [63,64]. Determining what qualifies a TMB-H tumor for each malignancy will help determine the usefulness of this strategy and predicting ICI response.

## 5. Epstein-Barr Virus (EBV)

Epstein-Barr virus (EBV) is a herpes virus and is linked to cancers in humans including nasopharyngeal carcinoma (NPC), GAC, and a variety of lymphomas (Burkitt’s lymphoma, Hodgkin’s lymphoma, and NK/T cell lymphoma) [65,66,67]. Approximately 10% of GACs are EBV positive [65]. EBV GAC is most found in the upper stomach and has a diffuse histology with lymphoid infiltrate and a higher overall survival [68,69]. By influencing host genome methylation and gene expression, the viral protein, EBV-encoded RNAs noncoding RNA, and EBV miRNAs contribute to carcinogenesis.

The mechanism by which EBV infects gastric epithelial cells is unknown, while EBV infection is common in B lymphocytes and the oral epithelium. It’s possible that EBV-infected saliva is swallowed, and the virus infects epithelial cells directly. Another theory is that EBV is reactivated in the stomach’s B lymphocytes and discharged to infect epithelial cells [70]. The lack of EBV infection in premalignant gastric lesions leads to the theory that the virus infection occurs later.

Only a small percentage of those infected with EBV form malignancies, showing that host cell molecular abnormalities are also significant in EBV-associated carcinogenesis. High-frequency mutations of phosphatidylinositol-4,5-bisphosphate 3-kinase catalytic subunit alpha (PIK3CA), at-rich interactive domain-containing protein 1A (ARID1A), and BCL6 Corepressor (BCOR) have been discovered in EBV GAC [17,71]. Higher levels of PD-L1 expression in carcinoma cells and infiltration of PD-L1 + immune cells are seen in EBV GAC. Avelumab, anti-PD-L1 agent, showed a remarkable effect in a patient with EBV GAC [72]. Additionally, a phase 2 trial with pembrolizumab in refractory (progression on 1–2 lines of chemotherapy) metastatic GAC (*n* = 61) [73]. Six cases had EBV GAC. ORR in these cases was 100 percent. Trials looking specifically at ICI therapy in EBV GAC are underway [74,75,76,77].

## 6. DNA Polymerase Epsilon (POLE) and Delta-1 (POLD1)

POLE and POLD1 are genes that encode DNA polymerases which are key enzymes for proofreading and fidelity of DNA replication [78]. They are essential in suppressing gene mutations and tumorigenesis. Mutations in POLE and POLD1 have been associated with a high TMB, deficient DDR, response to ICI, and an improved prognosis. Wang et al. reported on the prevalence of these mutations in patients with different cancers (*n* = 47,721) [78]. POLE/POLD1 mutations were seen in CRC (*n* = 197/2674 = 7%); gastroesophageal cancer (*n* = 185/2586; 7%), hepatobiliary cancer (*n* = 50/1759 = 2.8%), and pancreatic cancer (*n* = 25/1400 = 1.8%). In the same report, the authors noted that patients with either POLE or POLD1 mutations had longer OS with 34 months vs. 18 months for those that were wild type, *p* = 0.004. Garmezy et al. showed similar results in patients identified to have a POLE mutation (*n* = 458) [79]. Eighty-two patients received ICI therapy. Those with pathologic POLE mutations had improvement over those with benign variants (clinical benefit ratio 82.4% vs. 30.0%, *p* = 0.013; median PFS 15.1 months vs. 2.2 months, *p* = 0.001; and OS 29.5 months vs. 6.8 months, *p* = 0.001). The role of POLE and POLD1 mutations in GI tumors is far from being defined. We believe basket solid tumor trials with these mutations will be the short-term key to understanding what role these play in cancer medicine. Currently, ICI exploration is underway in MSI-H/dMMR or POLE mutations solid tumors [34,80,81,82].

## 7. DNA Damage Repair (DDR)

Genomic stability requires protection from DNA damaging agents [83]. A key mechanism for safeguarding intrinsic and extrinsic DNA damage is through DNA Damage Response (DDR). Examining the DDR system has led to targeted therapy in recent years. Two pathways involved in double-strand DNA break repair are homologous recombination repair (HRR) and non-homologous end joining (NHEJ). Breast cancer gene-1 (*BRCA1*) and breast cancer gene-2 (*BRCA2*) are players in the HRR machinery. The lack of functional *BRCA1* and *BRCA2* causes deficiency in HRR leading to defective double-strand DNA break repair. Frequency of these mutations in GI tumors are still being elucidated but appear at a wide range dependent on the mutation and GI tumor (i.e., *BRCA1* pancreatic cancer = 1.3–1.4%; *BRCA2* pancreatic cancer = ~3%; *BRCA1* biliary tract cancers = 1%; *BRCA2* biliary tract cancers = 2%; etc.) [84]. Poly–(ADP) ribose polymerase inhibitors (PARPi) have been developed which inhibit base excision repair at single-strand DNA break and lead to double-double strand DNA break [84]. As a result, “synthetic lethality” is induced in cancers exhibiting HRR deficiency, such as *BRCA* mutations [83,84].

Although the significant progression-free survival benefit was observed in the phase III POLO trial on active olaparib maintenance therapy versus placebo for patents with pancreatic cancer [83,84], combination strategies with ICIs and PARPi are underway. The rationale being that DDR defects are correlated with a higher neoantigen load and TMB. NCT04493060 and NCT04548752 are currently investigating this strategy in pancreatic cancer [85,86].

## 8. Gut Microbiota

The gut microbiota is composed of bacteria, fungi, protozoa, archaea, and viruses living in the GI tract [87]. It represents a complex ecosystem and plays a significant role in host digestion, nutrient absorption, metabolism, and immunity. The gut microbiome, in recent years, has been linked to the development of certain tumors. Additionally, microbiome changes have been shown to impact ICI and chemotherapy effects. We are only at the beginning stages of understanding how the microbiome relates to GI tumors. We anticipate much more research in this area such as microbiome etiology relation to certain cancer developments and certain treatments to be elucidated soon. One such impactful potential in early stages is the role of fecal transplants from ICI responders to enhance ICI tumor recognition in ICI non-responders. A small melanoma study looking at this showed six out of 15 patients responded or had disease stabilization with some having a long-term response [88]. Early correlation has been described in GI tumors [89,90] Trials are underway including exploring in those with MSI-H/dMMR non-ICI responsive disease [91,92]. We look forward to the answers given by this investigation.

## 9. Conclusions

Significant strides in recent years have identified biomarkers that can help predict the potential for ICI response in GI tumors. Unfortunately, these biomarkers (MSI-H/dMMR, PD-L1 overexpression, EBV, TMB, POLE) have been linked to potential ICI responses but not in a large fraction of GI tumors. A more thorough understanding is also needed as to why those with these certain rare biomarkers do not respond to ICI. More breakthroughs via translational medicine are needed to establish factors that make for “cold” tumors—those tumors unresponsive to ICI therapy. Additionally, how traditional markers such as human epidermal growth factor receptor-2 (HER-2) in GAC and Kirsten rat sarcoma viral oncogene (KRAS) in CRC impact ICI response need to be elucidated. We look forward to future discoveries in the role of ICI in GI tumors.

## Figures and Tables

**Table 1 cancers-14-04804-t001:** Microsatellite Instability-High/Deficient Mismatch Repair (MSI-H/dMMR) Phase 3 Trials in Gastrointestinal Malignancies [29,30,31,32,33,34,35].

Trial Identifier	ICI Therapy	Phase	Patient Population	Setting
NCT02997228	Atezolizumab +/− bevacizumab with chemotherapy	3	CRC	Metastatic
NCT04008030	Nivolumab +/− ipilimumab or chemotherapy	3	CRC	Metastatic
NCT05239741	Pembrolizumab vs. chemotherapy	3	CRC	Metastatic
NCT05236972	Sintilimab vs. CapeOx	3	CRC	Postoperative
NCT04304209	Sintilimab +/− chemotherapy	2/3	CRC	Preoperative/Watch and wait
NCT03827044	Avelumab + chemotherapy	3	Colon cancer	Postoperative
NCT05002686	Sintilimab + chemoradiation	2/3	Gastric cancer	Preoperative

CRC: colorectal cancer.

**Table 2 cancers-14-04804-t002:** Programmed-Death Ligand-l Expression with Immune Checkpoint Inhibitors in Gastric Cancer [36,37,38,39,40].

Trial Name/Identifier	ICI Therapy	Phase	Setting	Results
CHECKMATE-649 NCT02872116	Chemotherapy +/− nivolumab	3	Metastatic PD-L1 not inclusion criteria. Results reported by CPS score	Median OS: CPS ≥ 5: 14.4 months vs. 11.1 months CPS < 5: 12.4 months vs. 12.3 months Any CPS: 13.8 months vs. 11.6 months Median PFS: CPS > 5: 7.7 months vs. 6.0 months Any CPS: 7.7 months vs. 6.9 months
KEYNOTE-811 NCT03615326	Trastuzumab + chemotherapy +/− pembrolizumab	3	Metastatic PD-L1 not inclusion criteria	ORR: 74.4% vs. 51.9% Complete response: 11.3% vs. 3.1%
ATTRACTION-4NCT02746796	Chemotherapy +/− nivolumab	2/3	Metastatic PD-L1 not inclusion criteria Results not defined by CPS score (only ~15% in each group had PD-L1 expression ≥ 1)	Median OS: 17.45 months vs. 17.15 months Median PFS: 10.45 months vs. 8.34 months
JAVELIN Gastric 100 NCT02625610	Avelumab maintenance therapy vs. continued chemotherapy	3	Metastatic PD-L1 not inclusion criteria Results described by PD-L1 expression and CPS	Median OS: All patients: 10.4 months vs. 10.6 months PD-L1 ≥ 1% expression: 16.2 months vs. 17.7 months PD-L1 CPS ≥ 1: 14.9 months vs. 11.6 months
ORIENT-16 NCT03745170	Chemotherapy +/− sintilimab	3	Metastatic PD-L1 not inclusion criteria Results reported by CPS	Median OS: All patients: 15.2 months vs. 12.3 months CPS ≥ 5: 18.4 months vs. 12.9 months

PD-L1: programmed death ligand-1; OS: Overall survival; CPS: combined positive score; PFS: progression-free survival.

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
