# Peer review of "Current Immune Checkpoint Inhibitor Genetic Biomarker Exploration in Gastrointestinal Tumors"

_cancers, 2022, doi:10.3390/cancers14194804_

Round 1
Reviewer 1 Report
1. Keywords should be reduced to five, and abbreviations that appear for the first time should be clearly written in full.
2. Some references directly related to some sections are not listed. For example, for 8. Gut Microbiome, “The Gut Microbiome Is Associated with Clinical Response to Anti-PD-1/PD-L1 Immunotherapy in Gastrointestinal Cancer (PMID:32855157)” , “Ginseng polysaccharides alter the gut microbiota and kynurenine/tryptophan ratio, potentiating the antitumour effect of antiprogrammed cell death 1/programmed cell death ligand 1 (anti-PD-1/PD-L1) immunotherapy (PMID:34006584)” , “ Gut microbiome is associated with the clinical response to anti-PD-1 based immunotherapy in hepatobiliary cancers (PMID:34873013)”, “Gut microbiome modulates response to anti-PD-1 immunotherapy in melanoma patients (PMID: 29097493)” et al. were not cited.
3. Some references are not normative, such as Ref: 29-35, 43, 74-77, 80-82, 85-86, 89-90. It is suggested that clinical trials for each part were presented in tables in the manuscript. And clinical trials of immune checkpoint blockade in the gastrointestinal tract are not complete.
4. Summary for each part is recommended.
Reviewer 2 Report
General comment
The work provides an update on the current standing of the immune checkpoint biomarkers in GI tumors and their perspectives on the development of effective combination strategies for immunotherapy of GI tumors. The work is well structured, well written, easy to read, and provides a systematic and concise update of the topic in the field of cancer immunotherapy. The reviewed topic is current and is very likely to be of interest to readers in the field of cancer therapy. There is only one minor comment to the manuscript.
Minor revision:
1/ The review is nearly exclusively focused on the genetic-based biomarkers and does not provide updates on other types of biomarkers in GI tumors, such as tumor-infiltrating lymphocytes, their phenotypic profiling, or immunohistochemistry. Therefore, the title should acknowledge this focus and include, for instance, the word “genetic” in the title, i.e., “Current immune checkpoint inhibitor genetic biomarker exploration in gastrointestinal tumors.”
Reviewer 3 Report
Dear Sir,
The article Current immune checkpoint inhibitor biomarker exploration in gastrointestinal tumors by JE Rogers et at addresses a topic of interest, which is the search for markers to predict the effectiveness of checkpoint inhibitors in the treatment of digestive tumors. It is a short review that does not go into much detail but is useful to get a quick idea of the situation in this field. However, some aspects should be improved to clarify the text, which is somewhat confusing, mainly taking into account that, by its nature, a review should be useful for professionals who are not particularly familiar with the subject.
For example, some abbreviations are not explained (PMS2, HCC, EBER...) and others are explained several times (ICI in lines 39 and 87). Similarly, not all readers should be aware of what an agnostic tumor is, nor what Checkmate649, Keynote811, Javelin and the other assays are. In some parts of the text assays are taken for granted (sections 2 and 3) while in others it is specified what they are (section 4). Perhaps a table would make the results discussed in the text more visual.
Finally, I encourage the authors to revise the text including missing words (lines 45, 54, 60, 105), grammatical (lines 109, 298) and scientific errors (in section 8 they refer to the microbiota not to the microbiome) and some reference to the statement in line 308.
